# Prevalence of Musculoskeletal Manifestations in Adult Kidney Transplant’s Recipients: A Systematic Review

**DOI:** 10.3390/medicina57060525

**Published:** 2021-05-23

**Authors:** Adla B. Hassan, Kanz W. Ghalib, Haitham A. Jahrami, Amgad E. El-Agroudy

**Affiliations:** 1College of Medicine and Medical Sciences, Arabian Gulf University (AGU), 323 Manama, Bahrain; kanzalkinooz@hotmail.com (K.W.G.); hjahrami@health.gov.bh (H.A.J.); 2Department of Internal Medicine, University Medical Center (UMC), King Abdullah Medical City (KAMC), 323 Manama, Bahrain; 3Ministry of Health, 410 Manama, Bahrain

**Keywords:** bone pain, CyA-induced pain syndrome, renal transplant, osteoporosis, musculoskeletal, hyperuricemia

## Abstract

*Background and Objectives*: The musculoskeletal (MSK) manifestations in the kidney transplant recipient (KTxR) could lead to decreased quality of life and increased morbidity and mortality. However, the prevalence of these MSK manifestations is still not well-recognized. This review aimed to investigate the prevalence and outcomes of MSK manifestations in KTxR in the last two decades. *Materials and Methods:* Research was performed in EBSCO, EMBASE, CINAHL, PubMed/MEDLINE, Cochrane, Google Scholar, PsycINFO, Scopus, Science Direct, and Web of Science electronic databases were searched during the years 2000–2020. *Results:* The PRISMA flow diagram revealed the search procedure and that 502 articles were retrieved from the initial search and a total of 26 articles were included for the final report in this review. Twelve studies reported bone loss, seven studies reported a bone pain syndrome (BPS) or cyclosporine-induced pain syndrome (CIPS), and seven studies reported hyperuricemia (HU) and gout. The prevalence of MSK manifestations in this review reported as follow: BPS/CIPS ranged from 0.82% to 20.7%, while bone loss ranged from 14% to 88%, and the prevalence of gout reported in three studies as 7.6%, 8.0%, and 22.37%, while HU ranged from 38% to 44.2%. *Conclusions:* The post-transplantation period is associated with profound MSK abnormalities of mineral metabolism and bone loss mainly caused by corticosteroid therapy, which confer an increased fracture risk. Cyclosporine (CyA) and tacrolimus were responsible for CIPS, while HU or gout was attributable to CyA. Late diagnosis or treatment of post-transplant bone disease is associated with lower quality of life among recipients

## 1. Introduction

Kidney transplantation has been proven as a gold standard therapy for end-stage renal disease. The prevalence of musculoskeletal (MSK) manifestations, especially bone symptoms, is relatively high in kidney transplant recipients (KTxR) leading to decreased quality of life and increased morbidity and mortality. However, the precise prevalence of these MSK manifestations is still not well-recognized. These MSK manifestations have been reported in the forms of bone loss [1], joint pain [2], gouty arthritis [3,4], myopathies [5], and bone pain syndrome (BPS) of lower limbs [6]. Other studies reported them as transient osteoporosis, stress fractures, persistent hyperparathyroidism (pHPTH)-induced bone diseases, osteonecrosis, soft tissue infections, and osteomalacia [3,7,8,9,10]. Those post-KTx bone diseases occur in adults, as well as in children and adolescences [11,12].

The BPS of the lower limbs was first described in the 1990s and was related to the use of calcineurin inhibitors (CNI), such as cyclosporine A (CyA) to prevent graft rejection after organ transplantation [8,13,14,15], and later over the years, an increased incidence of BPS was also related to the tacrolimus [6,16,17,18]. The BPS was named calcineurin inhibitor-induced pain syndrome (CIPS) by Grotz et al. in 2001 [7]. BPS is a frequent complication that can be caused by several diseases and few medications include pHPTH, KTx malfunction, obesity, and treatment with glucocorticoids [13,19]. Although bone pain is a common post-KTx problem, acute inflammatory arthropathy is very rare [20]. The BPS or CIPS should be distinguished from a broad list of differentials that includes septic arthritis, crystal arthropathies, rheumatological disorders, and medication-induced [21]. It is well-known that Hyperuricemia (HU) or clinical gout in adult KTxR is more severe, involving unusual joints, with an early onset and fast progression of tophaceous lesions [4]. Some predictors for the development of HU in KTxR have been described [22]. In KTxR gout was less frequent than HU with a prevalence of 2–13% [22,23,24]. Whereas, HU has a prevalence of 19–55% in patients who treated with non-CyA therapy and from 30–84% in patients treated with CyA [22,25]. Bone loss associated with osteoporosis in KTxR is the greatest in the first 6–12 months. Bone loss in KTxR, which could be temporal, has been studied earlier and appeared to be less noticeable compared to bone pain. The KTxR may lose 6–7% of vertebral bone mineral density (BMD) within the first six months [9], but whether this was expressed in a higher incidence of fractures remains to be clarified [9,26,27], and the majority of post-KTx fractures are peripheral with an incidence rate exceeds 40%. A study from the United States revealed that the incidence of fractures in post-KTxR that led to hospitalization after a median follow-up of 32 months was higher in steroids recipients compared to non-steroid recipients, and the commonest fracture sites were femur, ankle, and spine [26]. Bisphosphonates, anti-osteoporosis therapy, were highly beneficial in the prevention of early bone loss after KTx [27]. High doses of corticosteroid and pHPTH are the most important factors inducing BMD reduction in long-term KTxR, if parathyroidectomy is required, in severe HPTH associated hypercalcemia, then subtotal is recommended. Additionally, active vitamin D with or without bisphosphonates use and steroid withdrawal are all effective in preventing early post-KTx bone loss [26,28,29,30].

In the current systematic review, we aimed to investigate the existing knowledge of the prevalence, severity, and outcomes of musculoskeletal symptoms in adult KTxR in the previous last two decades (2000–2020).

## 2. Materials and Methods

### 2.1. Data Search

Two reviewers performed the search independently. A search was performed in EBSCO, EMBASE, CINAHL, PubMed/MEDLINE, Cochrane, Google Scholar, PsycINFO, Scopus, Science Direct, and Web of Science electronic databases were searched during years 2000–2020. Only randomized controlled clinical trials (RCT) and observational/original studies in English languages were reviewed. We also manually checked the references of included articles for potential inclusion in this systematic review. This search resulted in the eligibility of some more studies of published articles.

Using the two basic Boolean operators “AND, OR” we did a comprehensive search on kidney transplant and musculoskeletal or rheumatic manifestations by group and by individual diseases, as follow; “Post kidney transplant” OR “Post Renal transplant” AND any of the following; “Bone pain syndrome”, “bone syndrome”, “bone pain”, “bone disease”, “lower limbs pain”, “distal limbs syndrome”, “musculoskeletal”, “gout”, “HU”, “ hyperuricemia”, “arthritis”, “bone loss”, “osteoporosis”, “osteopenia”, “persistent hyperparathyroidism”, “hypercalcemia”, “osteonecrosis”, “ AVN”.

### 2.2. Study Eligibility Criteria

The RCTs and original articles (prospective or retrospective, cross-sectional, or longitudinal observational studies) that involved human adults, English language, and directly studied musculoskeletal manifestations, which developed after the kidney transplant were included in this systematic review. Studies that involved participants of unisex or lack of a control or placebo group were not excluded.

Exclusion criteria included systemic reviews; meta-analysis; letters to the editor and case reports, including only one patient; studies that involved pregnant, or breastfeeding women; any article of the year (2000–2020); authors; or abstracts that were not available in the early searching process and; studies that involved participants with musculoskeletal affection before kidney transplants.

As a general overview, two reviewers participated in the initial selection of the studies according to our search strategy as in the PRISMA flow diagram (Figure 1). Two other independent reviewers assessed the papers according to our selection criteria and extracted the data. The four reviewers reviewed all study characteristics (Table 1, Table 2 and Table 3). The recruitment criteria for each study were briefly described, all studies included in this review showed a clear methodology and subsequent reporting of their results of prevalence (all studies described in Table 1, Table 2 and Table 3).

## 3. Results

Figure 1 shows the PRISMA flow diagram for this review. It reveals the search procedure and states that 502 articles were retrieved from the initial search. A total of 26 articles were included for the final report; 12 studies reported bone loss, 7 studies reported BPS/CIPS, and 7 studies reported HU and gout.

Table 1 depicts seven studies investigating BPS/CIPS from different countries: Two studies from Germany and one from each of Australia, Belgium, Turkey, France, and Italy. The prevalence of BPS or CIPS in the seven studies ranging from lowest to highest were as follows: 0.82%, 1.25%, 2.2%, 2.98%, 5.8%, 6%, and 20.7%.

Table 2 Depicted seven studies investigating gout and HU. Two studies from the United States, one study from each of Japan, Portugal, Taiwan, Mexico, and Latvia. The prevalence of gout reported in three studies was 7.6%, 8.0%, and 22.37%, while HU was reported as 38%, 42.1%, 42.4%, and 44.2%. One study reported both HU and gout. One of the 7 studies had selected a cohort and reported a prevalence of Monosodium Urate (MSU) crystal deposition of 0.03% among 27 patients with post KTx gout.

Table 3 depicts 12 studies investigating the bone loss. Low BMD (osteoporosis and osteopenia) reported in 6 studies, persistent HC due to persistent hyper-PTH reported in 3 studies, AVN or ONF in 2 studies, and osteomalacia in one study. The 12 studies from different countries; three from Japan, two from Unite states, two from Egypt, and one from Spain, Italy, Switzerland, France, and Australia. The prevalence reported by those studies as follow; low BMD (osteoporosis and osteopenia) reported in 6 studies as follows: 88%, 36.36%, 76.2%, 82.1%, 14.5% of the spine, and 3.2% in the pelvis. In one study, osteopenia and osteoporosis ranged from 5–35%at different sites. Hypercalcemia was reported in 3 studies as 21%, 47.1%, and 15%. The presence of osteomalacia was reported in one study as 52.7%. ONF/AVN reported in one study as 7.28% and another study as 33%, in addition to 54.1% enthesopathy (Gluteus minimums and Medius tendons abnormality).

## 4. Discussion

To our knowledge, this is the first systematic review to estimate the global prevalence of musculoskeletal (MSK) manifestations in adult kidney transplantation recipients (KTxR). The MSK manifestations in KTxR are the leading cause of decreased quality of life and increased morbidity and mortality, however, to date the exact prevalence of these complications is still not well-known. In the current review, we aimed to investigate the prevalence, severity, diagnosis, and outcomes of MSK manifestations among adult KTxR in the previous last two decades (2000–2020). In our present review, we found that the most common reported MSK manifestations in adults was the bone loss reported in twelve articles; three of them reported osteoporosis and osteopenia, another three reported persistent hypercalcemia (HC) because of persistent hyper-PTH, and two reported AVN/ONF. BPS or CIPS reported in seven studies, similarly, gout and HU were also reported in seven studies.

### 4.1. Discussion of the Studies Included in This Review: Bone Pain Syndrome (BPS)

The BPS of the lower limbs is characterized by its bilateral and symmetrical pattern of feet, ankles, or knees. The prevalence of BPS or CIPS in seven included studies was as follow: 0.82%, 1.25%, 2.2%, 2.98%, 5.8%, 6%, and 20.7% [2,7,17,31,32,33,34]. The prevalence of BPS or CIPS range from <1% to 6% in 6 studies and only about 20% in the Turkish study where only CyA was used. Interestingly, in two studies, only CyA was used. In another study, only two Tac were used, and in the remaining three studies both CyA and Tac were used.

A study by Grotz et al., 2001, revealed that eight patients developed severe BPS, mainly due to high doses or serum levels of CNI, such as CyA [7]. Although this syndrome was rare and accounted for around 1.3%, it is considered to be severe with a mean pain score of 71%. The CIPS is accurately diagnosed by its typical clinical presentation and radiological findings of bone scan and magnetic resonance imaging (MRI), which showed in the painful bone area as an increased tracer uptake and bone marrow edema, respectively [7]. These MRI findings have been reported by another author, Coates et al. in 2002, who described a prevalence of CIPS of 3%. The patients in that study presented with severe knee pain within three months post-transplantation. Plain radiographs and inflammatory markers were normal. In all cases, the MRI showed a distinctive pattern of bone marrow signal changes, where in some cases, extended from the epiphyseal region into the metaphyseal region during the follow-up period of 36 months. Recommendations were made not to withdraw cyclosporin, since the pain resolved spontaneously over three months without therapy, additional to the resolution of the MRI [31]. Michel Franco et al., 2003, investigated the role of tacrolimus in CIPS among KTxR and found that tacrolimus may induce the same bilateral symmetrical lower limbs pain syndrome (CIPS) as CyA does. They revealed that the CIPS has classic radiologic signs. Whereas, Technetium 99 m bone scanning showed increased uptake in the affected areas, and MRI changes were consistent with bone marrow edema [32]. These radiological signs were consistent with the previous study in reflex sympathetic dystrophy syndrome (RSDS), where the most difficult disorder is distinguishable from CIPS since it has a similar radiological picture. However, unlike CIPS, it has an asymmetrical distribution and characteristic trophic skin lesions [18]. In the same year, in 2003, Hamide Kart-Koseoglu et al., determined the prevalence of joint pain and arthritis in KTxR, and explored its relationships with CyA dose, as it was significantly correlated with serum levels higher than 200 ng/mL. Moreover, BMD analysis indicated that more than half of the studied patients (62.2%) showed osteopenia or osteoporosis, but their T-scores did not correlate with joint pain or arthritis [2]. Goffin, E et al. in 2003 documented both the occurrence of BPS in 6% of post-KxR given tacrolimus, but also the spontaneous resolution of the symptoms within a few weeks together with the marrow abnormalities similar to what has been attributed to CyA previously [17]. Later in 2006, A Collini et al. investigated the CIPS after KTx by performing different radiological measures and showed a low prevalence of 0.82%, a minimal amount of articular effusion, and a mild synovial reaction in the knees and feet joints by ultrasound. While, MRI showed an area of bone marrow edema in the external condyle of the femur and wearing of the cartilage. Moreover, a computerized bone mineralometry showed a slight reduction of the bone mass, while bone scintigraphy revealed increased radionuclide uptake in the affected joints. They stated that CIPS is rare, but has a distinct radiological feature and confirmed its reversibility over a period of a few months, without any sequelae [33]. Their radiological findings were consistent with another study [53], but their results of reversible and complete recovery of CIPS was inconsistent with another study that reported an association of CIPS with a stress fracture of the bone, and stated that both bone insufficiency and epiphyseal impaction may play a role [54]. A German study in 2008, by Frank-Peter Tillmann et al., determined the clinical diagnosis and long-term outcomes of post-transplant BPS, which is also known as post-transplant distal limb syndrome (PTDLS). They demonstrated a prevalence of 5.8% that typically developed within the first-year post KTx, and which may lead to significant morbidity because of pain induced immobilization [34]. Their results were consistent with other studies that stated a delay or incorrect diagnosis of this syndrome will lead to a significant reduction in life quality [8,13].

### 4.2. Discussion of the Studies Included in This Review: Hyperuricemia (HU) and Gout

In the years before 2000, gout was a common complication among KTxR with a prevalence of 2 to 13%, while HU was more prevalent and could account for more than 80% [55,56,57,58,59,60,61]. The prevalence of gout reported in three studies was 7.6%, 8.0%, and 22.37%, while HU ranged from 38% to 44.2%. The prevalence of disease in our studies was credited to CyA use, which is known to lower urinary clearance of uric acid by a mechanism not well-understood but could be attributed to some alterations in tubular transport [22].

Abbott. et al., in 2005 [35], found that the incidence of new-onset gout after KTx is 5.5%, while the cumulative incidence of new-onset gout at three years was 7.6%. The authors also determined that CyA was an independent risk factor for gout development as it induced gout in 6.0%, compared to tacrolimus, which was the least risk factor [35]. These results are consistent with another study reporting a lower incidence of new gout associated with tacrolimus use, compared to CyA [62], but are inconsistent with a study reporting no differences between both drugs in inducing new gout [63]. An association of new-onset gout with azathioprine, since it has known interactions with allopurinol [64]. Therefore, non-CyA treated KTxR was found to be 23%, where KTxR treated exclusively with azathioprine and prednisone [3]. Kevin C and his colleagues found that the new-onset gout was an independent predictor for death and graft loss [35], and their results were consistent with few previous studies, which reported that even with the management of uric acid levels, only mild improvements in GFR in KTxR could occur, which can contribute to the recovery of renal function in non-KTxR [4]. Kimura-Hayama et al., in 2014, studied the faster tophaceous monosodium urate crystals (MSU) deposition found that the prevalence of new-onset gout-induced tacrolimus in KTxRwas 70.4%, while it was 40.7% with CyA. They scanned 351 anatomical regions in the 27 KTxR cases with HU and identified only one patient with MSU deposition in the quadriceps tendon [39]. This finding was inconsistent with what was already known about the commonest sites for tophaceous gout MSU crystals deposition were the Achilles and the peroneal tendons [65]. Brighama M et al., 2019, compared the severity and treatment of gout in patients with or without history of KTx, they found that 8.0% of KTxR has gout. They demonstrated greater prevalence of severe uncontrolled gout of 27%, tophi 36% and higher rates of failure to allopurinol in gout patients with KTx compared to gout patients without KTx [40].

In the study by Malheiro J. et al., 2012, demonstrated that the prevalence of HU in KTxR was 42.1%. KTxR with HU were predominately male and older, with lower eGFR, hypertension, dyslipidemia, prednisolone use, and CyA use versus tacrolimus. However, their findings propose that modifiable immunosuppression options could play a role in the management of gout [37]. In the same year, almost the same predictors for the development of HU in KTxR have been described by another study performed by Numakura et al., 2012, who described the use of calcineurin inhibitors, male gender, impaired renal function, higher body weight, and pretransplant dialysis for more than 36 months. It also showed that the prevalence of HU at one year after transplantation was 38% [36]. Weng, Shu, et al., 2014 [38], demonstrated that gout prevalence was 22.37%, while the prevalence of HU was 44.20%. Survival analysis showed the HU group had poorer graft survival than the normal-uremic group after a 13-year follow-up. They stated that the two groups had significantly reduced allograft survival due to high serum uric acid, which seems to be implicated in elevation of serum creatinine, and thus, allograft loss [38]. Recently, Inese Folkmane et al., in 2020, reported that the prevalence of HU in KTxR was 42.36% and its predictors included the presence of cystic diseases, the use of diuretics, and the male gender. Therefore, being a younger, female, with a normal BMI increased the possibility of normal GFR [41].

In the current review, the incidence of new-onset gout after KTx has not been specified per year at risk. Although, most of the data represented observational periods of five years or less. In many studies, which were mostly from a single center, the duration of the study was not standardized, thus, the rates or incidence from those different studies cannot be compared 100%.

### 4.3. Discussion of the Studies Included in This Review: Bone Loss

Bone loss in KTxR appeared to be more noticeable compared to BPS and gout or HU, where the prevalence of bone loss, almost within the first 12 months, ranged generally from 14% to 88%, including osteoporosis, osteopenia, osteomalacia, and AVN/ONF. KTxR may lose 6–7% of vertebral BMD within the first six months [9]. The prevalence of osteoporosis in long-term KTxR ranges between 11–56% with the incidence of vertebral fracture 3–29% and peripheral fracture 11–43% [24]. Bone loss occurs at its highest rate in the first six months after transplantation and continues to occur at a slower rate during the following 6–12 months [66,67]. The rate of bone loss in the first six months ranges between 5.5–19.5%, which decreases to 2.6–8.2% after 6–12 months. After the first year, BMD largely stabilizes, but in some patients, a gradual decline may still be observed at a rate between 0.4–4.5% [68]. Understanding whether bone loss was expressed at a higher incidence of fractures remains to be clarified [9,26,27]. Bone loss is the most pronounced in cancellous bone, while the cortical BMD is less influenced [67]. This pattern is similar to steroid-induced osteoporosis [66,69].

Antonio V. Cayco et al., 2000, has examined the prevalence of osteoporosis in long-term, more than one year, in 69 KTxR with preserved renal function, and BMD was measured using DEXA scan. They demonstrated that overall, 88% of the participants had reductions in bone mass at either the hip or the lumbar spine [43]. Monier-Faugere, M. et al., 2000, showed that generalized or focal osteomalacia was a frequent histologic feature in KTxR with bone pain, low bone volume, low bone turnover, or fractures. The effects of age, gender, PTH, and CyA on bone volume and bone turnover were ignored by the prominent effects of glucocorticoids. The prevalence of mineralization defect in the presence of normal serum levels of calcidiol and calcitriol suggests vitamin D resistance [42]. Ulivieri F, et al., 2002 showed that in twenty KTx male patients received one of two regimens (CyA combined with methylprednisolone with or without azathioprine). Serum PTH, phosphate, and calcium levels showed the expected decreasing trend up to six months after KTx as the normalization of renal function achieved. In male patients who underwent hemodialysis, there was a marked increase in fat mass, a significant loss of trabecular bone mass, and no change in cortical bone and lean mass [45]. Casez, J. et al., 2002, has investigated the role of prednisone and PTH in a selected cohort followed prospectively for 18 months after KTx. All patients received prednisone and CyA with, or without, azathioprine. The study revealed that high cumulative prednisone doses are deleterious for the axial skeleton and that the low levels of PTH observed during the first week after kidney transplantation is predictive of continuous cortical bone loss [44]. Inoue S, et al., 2003 demonstrated that among KTxR the only factors with relevance to the occurrence of ONF (6.27%) in such patients were the daily oral steroid dosage (25.0 mg/day or more) and blood urea nitrogen level two months after transplantation. They proposed that oral steroid dosages should be low or reduced after renal transplantation, and acute rejection should be controlled with pulsed therapy [47]. El-Agroudy et al. in 2003 suggested that early bone loss occurred during the first-year post KTx could be prevented by vitamin D metabolites and vitamin D receptor activators (alfacalcidol) with concomitant calcium supplements and that BMD after therapy increased by 2.1%, 1.8%, and 3.2% at the lumbar spine, femoral neck, and forearm, respectively and serum iPTH level decreased significantly after the therapy. Besides, osteopenia occurred at a prevalence of 35% at the lumbar spine, 30% at the femoral neck, and 25% at the forearm, whereas osteoporosis occurred at these sites by 10%, 15%, and 5%, respectively. Unfortunately, the overall prevalence of low BMD was not mentioned [48]. Michio Nakamura, et al., 2013, revealed that the persistent hyper-PTH-associated hypercalcemia during the early period in KTxR tends to persist for several years despite good kidney function due to remaining nodular hyperplasia, even if the glands are small. In such cases, it is desirable to perform the parathyroidectomy just before transplantation [28]. J Toro et al., in 2003, evaluated osteoarticular pain in KTxR and reported that the incidence of BMD was 76.2%, and found a significant incidence of fractures (frequently in vertebral column) mostly among post-menopausal women. A reduction in BMD values was associated with age, duration post-transplantation, body weight, and elevated PTH levels. There was a high risk of fracture in KTxR, which was more frequent in the vertebral column. In 23.8% of patients reporting osseous pain, there was no reduction in BMD levels, thus, other additional causes could be responsible for the pain [46]. The incidence of fracture after KTx in patients who received steroid was less in the spine (11%) compared to the femur (29%) and ankle (15%) and corticosteroid withdrawal were associated with a 31% reduction in the fracture risk. According to the above data, steroid withdrawal can preserve bone mass, especially in the central skeleton [26,70].

An Egyptian study by Atallah A et al., 2008, demonstrated that 117 KTxR were subjected to joint examination, including joint pain and morbidity measurements, but also BMD and Laboratory measurements. There were 81.2% complaining of MSK manifestations. These included bone loss of 82.1%, joint pain of 66.32%, BPS of 7.37%, but also skeletal muscle affection (*n* = 21), and soft tissue affection (*n* = 25) [50].

Any pre-and post-transplant factors increase bone loss and fragility, thus, intensify the fracture risk in KTxR [49]. The pre KTx factors are renal osteodystrophy, dialysis, age, gender, ethnicity, smoking, and BMI. On the other hand, the post KTx factors are steroids, high PTH, and low vitamin D, both steroids therapy and hyper-PTH lead to trabecular and cortical bone loss. Hyper-PTH, together with low vitamin D, led to abnormal bone mineralization and formation. Many other studies revealed that the risk factors involved low BMD post-KTx include increasing age, time after transplant, gender, and ethnicity [71,72,73,74]. Another study showed that the effects of age, gender, PTH, and CyA on bone density and bone turnover are masked by the prominent effects of steroids [42]. However, Aroldi and colleagues investigated the effects of immunosuppressive regimens on vertebral BMD in KTxR and revealed that the lumbar BMD decreased significantly in KTxR given CyA with steroids compared to those under CyA alone [75]. Steroids have been shown to reduce bone turnover and formation, as well as increase bone loss [76]. While late steroid withdrawal improves the BMD [77], an early steroid withdrawal minimizes bone loss [30], and reduced fracture risk by 31% [26]. Bone loss is most pronounced in cancellous bone, while the cortical BMD is less influenced [67]. This pattern is similar to steroid-induced osteoporosis [78]. The treatment of post-KTx bone loss should begin earlier than post-transplantation or before the transplant takes place [19], and can be prevented after KTx by alfacalcidol, calcitonin, or alendronate [79]. The first, conducted systematic review in bone loss in 2005, showed beneficial effects of bisphosphonates on BMD at the femoral neck and lumbar [80].

### 4.4. Persistent Hyperparathyroidism

Persistent hyperparathyroidism (pHPTH) with PTH of more than 130 ng/l is a risk factor for fractures, due to significant bone loss or low BMD as a result of HC [81]. In the study by Hiroo Kawarazaki and his group, a prospective study was performed involving Japanese KTxR investigating the pattern of PTH and calcium. Serum calcium levels increased until the fourth week post-KTx, after which it reached a plateau, and hyper-PTH persisted for 12 months after KTx. Their analysis revealed that hyper-PTH was the best correlating factor with persistent HCat 12 months in post-KTxR [51]. These results were consistent with another recent study that stated PTH could be normal in 20–30% at one year and could be high in 70–80% leading to hypercalcemia in 30–50% of patients with high PTH [81].

One of the interventions that could be recommended for better management of low BMD post KTx is to treat the pHPTH by parathyroidectomy or cinacalcet use [51,82]. A patient with pHPTH post-KTx should be treated if he or she has hypercalcemia (calcium > 2.8 mmol/L) either with active vitamin D or cinacalcet. Additionally, they could be treated if they have a significant bone loss of low BMD or fractures, similarly to those with high bone turnover or high bone alkaline phosphatase. Cinacalcet corrects mineral abnormalities in HPTH in the first 6 months post-transplant [76]. Both active vitamin D and paricalcitol suppress PTH post-transplant [83].

Successful KTx leads to a decline in PTH especially during the first 3–6 months after KTx. However, elevated PTH level can still be found in 30–60% at one year following transplantation and is associated with poor outcomes including osteoporosis, fracture, vascular calcification, and graft loss. For the first time, a study by Perrin, P. et al. in 2013 demonstrated that pHPTH is an independent risk factor for fractures, which occurred in 15.38% in the first five years after KTx [84]. A study by Amin T et al. in 2016 investigated the pHPTH in KTxR with a functioning allograft during 40 years follow-up. The prevalence of hypercalcemia was found to be 15% [52]. The prevalence of pHPTH post KTx has been generally appreciated earlier in 1992 using hypercalcemia as an index [9]. Secondary HPTH usually improves between the first month and six months post KTx with a reduction in parathyroid mass [85]. Risk factors for pHPTH include long dialysis duration, post-KTx HC, high alkaline phosphatase, and impaired kidney function. In KTxR with tertiary HPTH recurrence of HC after five-years follow-up is more frequent in cinacalcet than after subtotal parathyroidectomy [86]. Hypercalcemia in KTxR could be secondary to recovered circulating levels of 1,25 (OH)2 vitamin D due to increased renal tubular synthesis [87,88]. 

The present systematic review has some limitations. Besides being retrospective, some reports lack important data regarding graft survival and rejection. However, all reports lack data known to identify the risk factors for the development of MSK in general and could also play a role in KT patients, such as vitamin D serum levels and all bone health regulators, such as calcium, phosphorus, and alkaline phosphatase. Other limitations include that studies are very heterogeneous in their design and difficult to have one single conclusion apart from the prevalence. However, for the first time, we present data from a reasonably big-sized KT population all over the world, with a long follow-up time, and our main results are supported by old and more recently published data.

## 5. Conclusions

This review identified the full spectrum of musculoskeletal complications, for the first time, in this steadily increasing kidney transplant population. The post-transplantation period is associated with profound abnormalities of mineral metabolism, bone loss, and fragility, which confer an increased fracture risk. Significant challenges remain in the screening and diagnosis of post-transplant bone loss. Taken together, osteoporosis and osteonecrosis are mainly caused by corticosteroid therapy. Osteoporosis in our review had a prevalence that could reach 90%, osteoporosis leads to peripheral and vertebral fractures; osteonecrosis predominantly leads to early and severe osteoarthritis of the hip. Unawareness or treatment of post-transplant bone disease given too late is likely to reduce the life quality of the graft recipients and could lead to irreversible long-term complications. Joint or bone pain is an important problem in individuals with renal disease and is common both before and after KTx. CyA and tacrolimus are responsible for CIPS in all seven studies and have a prevalence range of 0.82% to 6% in six studies, but 20.7% in only one study. CIPS lowers the quality of life if the diagnosis is delayed. Gout is a common problem among renal transplant recipients, but HU is even more common than clinical gout. Their prevalence is attributable to CyA use, but individual patients may have other risk factors as well.

## Figures and Tables

**Figure 1 medicina-57-00525-f001:**
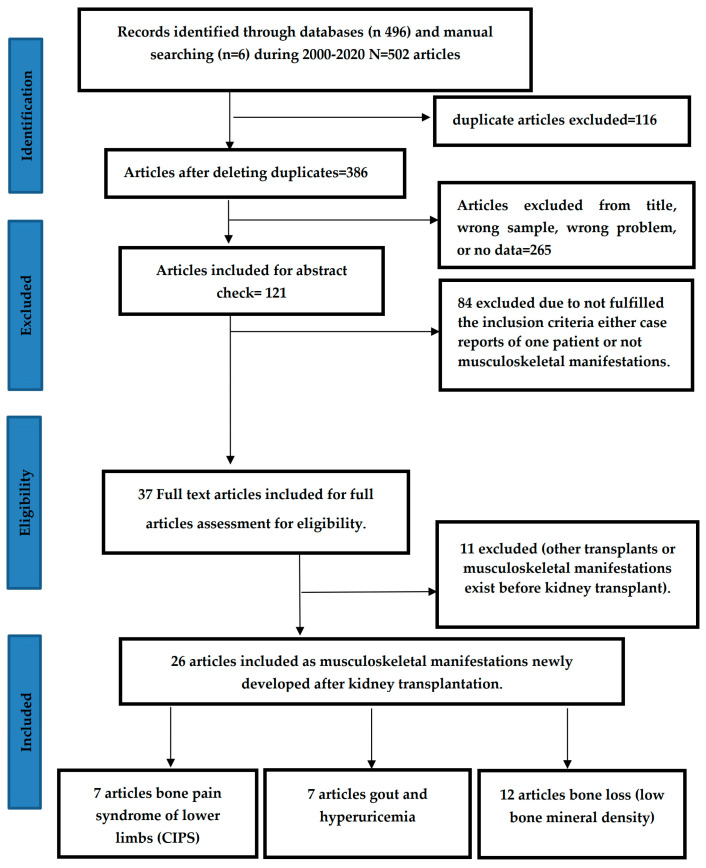
PRISMA 2020 Flow Diagram for newly developed musculoskeletal manifestations in adult post kidney transplant Recipients (KTxR).

**Table 1 medicina-57-00525-t001:** Prevalence of Bone pain syndrome (BPS) in adult post kidney transplant recipients (KTxR) included in this systematic review.

SN	1	2	3	4	5	6	7
Author et al. year	Grotz WH et al. 2001 [7]	Coates PT et al. 2002 [31]	Goffin E et al. 2003 [17]	Kart-Koseoglu H et al. 2003 [2]	Franco M et al. 2004 [32]	Collini A et al. 2006 [33]	Tillmann F. et al. 2008 [34]
Country	Germany	Australia	Belgium	Turkey	France	Italy	Germany
Type of study (MC/SC)	Prospective cohort study SC	Prospective cohort study SC	Prospective cohort study SC	Retrospective cohort study SC	Prospective cohort study SC	Retrospective cohort study SC	Retrospective cohort study SC
Objective	To investigate CIPS in the feet of 8 out of 637 KTxR using bone scan and MRI	To investigate MRI findings in 4 out of 134 KTxR who developed severe bilateral knee pain	To investigate the occurrence of the post-KTx syndrome in KTxR given a tacrolimus-based therapy	To determine the prevalence of joint pain and arthritis in KTxR	To investigate tac pain syndrome in 2 out of 90 KTxR on Tac over 3 years.	To investigate CIPS after KTx in 2 out of 243 pts	To determine the clinical diagnosis and long-term outcome of BPS of lower limbs in 37 out of 639 KTxR
T No of KTxR pts F: (%) M: (%)	8 F: 2 (25%) M: 6 (75%)	134 F: 2 (1.49%) M: 2 (1.49%)	86 F: 2 (40%) M: 3 (60%)	82 F: 26 (31.7%) M: 56 (68.2%)	2 F: 1 (50%) M: 1 (50%)	243 F: 0 M: 2 (0.82%)	37 F: NA M: NA
Musculoskeletal (MSK) manifestations Prevalence, No. (%) Time after KTx, M ± SD (range); months	CIPS 3 (1.25) NA (1–18)	BPS 4 (2.98) 1	BPS 5/(6) 6	BPS 17 (20.7) BMD done 51/82 (spine: osteopenia 39.2%, osteoporosis 31.4% and femur: osteopenia 47.1%, osteoporosis 31.4%) 12	CIPS 2 (2.2) NA (1–3)	CIPS 2 (0.82) NA (1–6)	BPS 37 (5.8) 5.8 ± 4.8 (1–30)
Age M ± SD years. (range) years.	NA (26–52)	46.75 (40–50)	46.6 (30–58)	18.41 ± 15.94 (12–56)	57 years (F) 49 years (M)	42 and 49	50.4 ± 10.9 (30.6–66.8)
MDD/DD Study period	NA 7 years	NA 3 months	22 months 10 moths	14 years 1 year	3 years 13 years	NA 15 years	5.1 ± 3.1 months 8 years
Type of Donor transplant	NA	Cadaveric	Cadaveric	18 cadaveric 60 first-degree relative 4 spouse	NA	Deceased	28 cadaveric 9 living
Causes of ESRD	IgA N, NPS, APKD, GS, Mesanigo-PGN, Reflux N Membrano-PGN,	-IgA Nephropathy -AN	HTN, PKD, NAS, CP, Unknown.	NA	NA	NA	NA
Pretransplant dialysis No. (duration in months)	3 PD (3–120) 5 HD (7–168.)	NA	1 PD (45) 2–5 HD (8–45)	NA 18	1 (28) 1 (29)	NA	NA
Immunosuppressive Therapy	CyA Tac CS MMF Aza ATG	CyA CS MMF Sirolimus	Tac CS MMF	CyA CS Aza	Tac CS MMF Aza	CyA Tac CS MMF Basiliximab Everolimus	CyA Tac MMF
Other therapy	Ibuprofen, Morphine, Alpha-lipoic, Carbamazepine, CCB Bisphosphonate,	NA	PCM	NA	Calcium Vit D Salmon Calcitonin Pamidronate	CCB	CCB
Renal function (Cr.; mg/dL); Mean (range)	NA	NA	5 ESRD 1.45–2.95	NA	NA	NA	1.79 ± 0.68 (0.80–4.40)
Graft (rejection)	NA	NA	1 Acute rejection 2 Chronic rejection	9 acute rejection	NA	1 acute rejection	NA
Co-morbidity No. (%)	NA	SLE	HTN	HCV Ab positive: 15 (18.3%)	NA	DLP: 1 (50%)	NA

Pts = Patients. years. = years. MDD = Mean disease duration. NA = Data not available. KTxR = Kidney Transplant Recipient. MC = Multicenter. SC = Single center. BPS = Bone pain syndrome. MSK = Musculoskeletal manifestations. CIPS = Cyclosporine-induced pain syndrome. HTN = Hypertension. ESRD = End-stage renal disease. NAS = Nephroangiosclerosis. CP = Chronic pyelonephritis. NPS = Nail-patella-syndrome. PGN = Proliferative Glomerulonephritis. PKD = Polycystic Kidney Disease. GS = Glomerulosclerosis. AN = Analgesic Nephropathy. HD = hemodialysis. PD = Peritoneal dialysis. MMF = Mycophenolate mofetil. CyA = Cyclosporine. Tac = Tacrolimus. Aza = Azathioprine. ATG = Anti lymphocyte globulin. CS = corticosteroids. PCM = Paracetamol. CCB = Calcium Chanel Blocker. SLE = Systemic lupus erythematosus. DLP = Dyslipidemia. MRI = Magnetic resonance imaging. HCV = Hepatitis C virus. Ab. = Antibody.

**Table 2 medicina-57-00525-t002:** Prevalence of hyperuricemia and gout in adult post kidney transplant recipients (KTxR) included in this systematic review.

SN	1	2	3	4	5	6	7
Author et al., Year	Abbott K et al. 2005 [35]	Numakura K et al. 2012 [36]	Malheiro J et al. 2012 [37]	Weng S.C et al. 2014 [38]	Kimura-Hayama E et al. 2014 [39]	Brigham M et al. 2019 [40]	Folkmane I et al. 2020 [41]
Country	United States	Japan	Portugal	Taiwan	Mexico	United States	Latvia
Type of study (MC/SC)	Retrospective cohort study MC	Prospective cohort study SC	Retrospective cohort study SC	Prospective cohort study MC	Prospective cohort study SC	Retrospective cohort study MC	Retrospective cohort study SC
Total No of KTxR (%) F: (%) M: (%)	28,942 (97.7%) F: NA M: NA	121 HU = 46 F: 8 (17.3%) M: 38 (82.6%) NU = 75 F: 37 (49.3) M: 38 (50.6%)	302 F: 119 (39.4%) M: 183 (60.5%)	880 HU = 389 F: 144 (37%) M: 245 (62.9%) NU: 491 F: 268 (54.5%) M: 223 (45.4%)	27 F: 20 (74%) M: 17 (63%)	312 (25 with Gout) F: 7 (28%) M: 18 (72%)	144 F: 111 (77%) M: 33 (22.9%)
Musculoskeletal (MSK) manifestations No. (%) Time after KTx, months	Gout: 1593 (5.5%) 7.6% (Cumulative at 3 years) 36	HU 46 (38%) 12	HU: 127 (42.1%) 90 (2.3–14.2)	HU: 389 (44.2%) Gouty: 22.37% 1–3	Gout (selected 27) Prevalence of MSU deposition is 0.03% 0.5–16.8 years.	Gout: 25 (8.0%) 12 months	HU: 61 (42.4%) ≥12 months
Age; M ± SD (range) years.	45.4 ± 14.6 (NA)	44.9 (20–78)	49.6 ± 13.4 (NA)	HU: 50.03 ± 12.07 (NA) NU: 47.59 ± 12.57 (NA)	44.7 ± 12.9 (NA)	55.1 ± 14.1 (NA)	46.6 ± 13.9 (NA)
MDD/DD Study period/years	NA 5 years	NA 5 years	NA 27 years	43.3 ± 26.3 months 14 years	3.2 years. 7 years.	NA 1 year	NA 3 years
Type of Donor transplant	Deceased	Allograft	Allograft	NA	NA	NA	NA
Total No Pts with ESRD Pts No (%) for diseases as causes of ESRD	59,077 ESRD 7145 (28.2%) DM 5312 (21.4) HTN 1262 (7.6) IHD 1872 (11.3) HF	121 ESRD: 70 IgA Nephropathy, 10 DN, 8 PKD, 4 Lupus nephritis, NS Alports syndrome, RN, 1 WG, 4 pregnancy toxicosis	302 ESRD	880 ESRD	27 ESRD	312 ESRD	144 ESRD: 17 PKD
Pretransplant dialysis No. of patients/% (Periods in months/years.)	NA	NA 56.3 (0–420) months	NA	HD: 29.56%	1 (0.5–10.6) years	NA	NA
Immunosuppressive Therapy used	CyA Tac MMF Aza Sirolimus	Tac CS MMF Basiliximab	CyA Tac CS MMF Aza Sirolimus	CyA Tac CS MMF Aza	CyA Tac MMF Aza Sirolimus	NA	CyA Tac CS MMF
Other therapy	NA	Allopurinol, MP pulse, Furosemide, Beta-blocker	ACEi	Allopurinol, Dithiazide, Benzbromarone, Aspirin	Allopurinol	Allopurinol, Febuxostat	ACEi, ARBs, Diuretics, Allopurinol Febuxostat
Renal function: mean Cr/CrC ± SD (mg/dl) eGFR (mL/1.73 m)	Cr. 1.8 ± 20.8 eGFR at 1 year post KTx < 44	NA eGFR: 47.6 ± 11.8	NA eGFR: 51.9 ± 18.46	Cr.: 2.96 ± 3.20 NA	CrC: 66 (7–119) NA	NA eGFR 45–59 mL/min: 65% eGFR < 15 mL/min: 6%	NA eGFT > 60: 19 (35.2%) at 3 years eGFR < 60: NA
Graft rejection Acute or chronic	Chronic (with the first year)	91.6% 5-year graft survival rate and decreased thereafter	Chronic with 43 (14.2%) pts had 2nd graft	60.47% poor graft survival	NA	NA	NA
Co-morbidity No. (%) * = with new onset gout	DM: 237 (3.5%)* HTN: 389 (7.3%)* IHD: 85 (6.7%)* HF: 117 (6.3)*	DM: 33 HTN: 7 DLP: 26 CMV: 10	HTN 246 (81.5%) DLP 208 (68.9%) DM 41 (13.6%)	HTN (92.29%), DM (140 (35.99), CVD (56 (14.40), DLP (62.98%), HCV, HBV (24.42%), TB, CMV, Shingles	HTN 24 (88.9%) DM 4 (14.8%) DLP 21 (77.8%)	NA	HTN: 51 (83.6%) DM: 7 (11.5%) DLP: 38 (62.3%)

Pts = Patients. years. = years. MDD = Mean disease duration. NA = Data not available. KTxR = Kidney Transplant Recipient. MC = Multicenter. SC = Single center. MSK = Musculoskeletal manifestations. MSU = Monosodium urate. DM = Diabetes Mellitus. HTN = Hypertension. DLP = Dyslipidemia. ESRD = End-stage renal disease. DN = Diabetic Nephropathy. PKD = Polycystic Kidney Disease. RN = Reflux Nephropathy. NS = Nephrosclerosis. HD = hemodialysis. MMF = Mycophenolate mofetil. CyA = Cyclosporine. Tac = Tacrolimus. Aza = Azathioprine. CS = corticosteroids. CVD = Cardiovascular Disease. IHD = Ischaemic hear disease. HF = heart failure. DLP = Dyslipidemia. HU = Hyperuricemia. NU = Normouricemia. Cr. = Creatinine. CrC = creatinine clearance. eGFR = Estimated glomerular filtration rate. CMV = Cytomegalovirus. TB = Tuberculosis. ACEi = Angiotensin-converting enzyme inhibitors. ARBs = Angiotensin II receptor blockers. MP = Methylprednisolone. HCV = Hepatitis C virus. HBV = Hepatitis B Virus. WG = Wegener granulomatosis.

**Table 3 medicina-57-00525-t003:** Prevalence of Bone loss in adult post kidney transplant recipients (KTxR) included in this systematic review.

SN	1	2	3	4	5	6	7	8	9	10	11	12
Author et al., Year	Monier-Faugere M et al. 2000 [42]	Cayco A et al. 2000 [43]	Casez JP et al. 2002 [44]	Ulivieri FM et al. 2002 [45]	Toro J et al. 2003 [46]	Inoue S et al. 2003 [47]	El-Agroudy A et al. 2003 [48]	Demant AW et al. 2007 [49]	Atallah A M et al. 2008 [50]	Kawarazaki H et al. 2011 [51]	Nakamura M et al. 2013 [28]	Amin T et al. 2016 [52]
Country	USA	USA	France	Italy	Spain	Japan	Egypt	Switzerland	Egypt	Japan	Japan	Australia
Type of study (MC/SC)	Prospective cohort study SC	Prospective cohort study SC	Prospective cohort study SC	Prospective cohort study SC	Prospective cohort study SC	Prospective cohort study SC	Prospective cohort study SC	Prospective cohort study SC	Prospective cohort study SC	Prospective cohort study SC	Prospective cohort study MC	Retrospective cohort study SC
Objective	To investigate the prevalence of bone disease in 57 out of 120 pts post KTx	To assess the prevalence of osteoporosis after one yr. post-KTx	To investigate the role of PDN and PTH in changing BMD following KTx	To investigate the effect of KT on bone mass and body composition in males	To evaluate the BMD and osteoarticular pain in adult patients who have undergone KTx	To investigate RFs for osteonecrosis of the femoral head in KTx among 18 out of 287	To investigate the effect of treatment with active Vitamin D on the prevention of Post KTx bone loss	To evaluate the diagnosis on MRI and radiography of KTxR with hip pain suspicious for AVN and symptomatic gluteal tendons abnormality	To determine Musculoskeletal affections (MSK) among KTxR and find possible risk predictors.	To show the natural history of mineral metabolism in post-KTxR and to clarify RF of persistent HC and hypo-phosphatemia at 12 mo after KTx	To investigate persistent HC and HPT post KTx in long-term dialysis	To determine the prevalence of HC and to evaluate the RFs for post-KTx HC in long-term KTxR
T No of KTxR pts F: (%) M: (%)	120 F: 25 (43.8) M: 32 (56.1)	69 F: 29 (42) M: 40 (58)	33 F: 19 (57.5) M: 14 (42.4)	20 F: 0 M: 20 (100)	123 F: 72(58.5) M: 51(41.4) 32.5% postmenopausal and 26% premenopausal	18 F: 7 (38.8) M: 11 (61.1)	40 F: 0 M: 40	24 F: 16 (66.6) M: 8(33.3)	117 95 (81.2%) with MSK symptoms F: 22(23.1%) M: 73 (76.8%) 22 (18.8%) without MSK F: M = 3: 19	34 F: 12(35) M: 22(65)	34 F: 12 (35) M: 22(65)	679 F: 39 (39) M: 61 (61)
Musculoskeletal (MSK) manifestations Prevalence, No. (%) Time after KTx M ± SD (range); months	Osteomalacia 57 (52.7) 60.4 ± 11 (6–247)	Osteoporosis and osteopenia 60 (88) Osteoporosis spine or hip 31 (44) Osteopenia 31 (44) >12	Low BMD (Whole body) 12 (36) 18	Trabecular bone loss of spine 3 (14.5) pelvis 1 (3.2) 6	BMD was reduced in 76.2% Osteopenia: 67 (54.8) Premenopausal 25 (37.5) Postmenopausal Men 42 (62.7) Osteoporosis: Premenopausal 16.1%, postmenopausal 50%, men 7.8% >12	ONF 7.28% (18) 0–12	Baseline osteopenia 35% at the lumbar spine, 30% at the femoral neck, and 25% at the forearm, whereas osteoporosis occurs at these sites (10%, 15%, and 5%, respectively 6–12	33% (8) AVN/ONF 54.1% (13) enthesopathy: Gluteus minimums and Medius tendons abnormality two pts has dual findings 20.8% (5) Normal findings > 6 mo	95 (81.2) Bone loss: 78 (82.1) Joint pain: 63 (66.3) Skeletal muscle affection: 21 (22.1) Soft tissue affection: 25 (26.3) BPS: 7 (37) 50 (1–242)	HC 21% Persistent hyper-PTH 100% (34) causing persistent hypercalcemia (21%) and hypo-phosphatemia (15%) 12	HC 47.1% (16) 12	HC 15% (101) March 2011-June 2011 3
Age MA ± SD (range) years	45 ± 2 (NA)	45 ± 11.2 (NA)	46 ± 2 (NA)	40 (23–64)	49.3 ± 9.87 (NA)	34.7 (20.5–56.6)	31.4 ± 10.1 (18–50)	57.1 (26–71)	38.6 ± 11.1 (19.6–56.0)	50 (36–60)	53.8 ± 7.9 (NA)	55± 13 (NA)
MDD/DD Study period, mo	NA NA	>12 NA	NA 18	109 ± 74 NA	NA 1980–2000	NA 1983–1992	NA 12	180 1998–2002	NA 2005–2006	12 2007–2008	162 ± 49 2002–2010	480 1971–2011
Type of Donor transplant	NA	41 cadaveric 28 living	NA	Living	Cadaveric	3 dead 15 living	Living	NA	NA	Living	Deceased	NA
Causes of ESRD	NA	NA	NA									34 CGN 16 RN 12 DN 12 PKD
Pretransplant dialysis No. (duration in months)	51 (36 ± 0.6) 32 HD 19 PD	1.74 ± 3.03	HD: 30 PD: 4 (63 ± 12)	HD: NA (23 ± 19)	NA	NA	NA (10.8 ± 5.4)	NA	NA (12)	NA	NA (172.8 ± 51.6)	65 HD 18 PD
Immunosuppressive Therapy	CyA CS MMF Aza	CyA Tac CS	CyA CS Aza	CyA CS Aza	NA	CyA Aza	CyA CS	CyA CS	CyA Tac CS Sirolimus	CyA Tac CS MMF	NA	NA
Other therapy	Diuretics Phosphate supplement	ERT	Phosphate supplements beta blockers CCB	NA	NA	-CS	NA	NA	NA	Active Vitamin D3 Phosphate Binders (Ca-containing)	NA	NA
Renal function (Cr.;mg/dl); Mean (range)	1.6 ± 0.1 (0.7–4.3)	1.4 ± 0.4	NA	NA	1.47± 0.8	NA	1.4 ± 0.4	NA	NA	NA	eGFR 47.2 mL/min	eGFR 30–60 mL/min
Graft (rejection)	NA	NA	13 AR	NA	NA	NA	NA	NA	15 CR	NA	NA	NA

Pts = Patients. years. = years. mo = months. MA = mean age. F: M = Female: Male. MDD = Mean disease duration. NA = Data not available. KTxR = Kidney Transplant Recipient. MC = Multicenter. SC = Single center. BMD = Bone mineral density. BPS = Bone pain syndrome. MSK = Musculoskeletal manifestations. RF = Risk factor. AVN = Avascular necrosis. ESRD = End-stage renal disease. PTH = Parathyroidism. CGN = Chronic Glomerulonephritis. DN = Diabetic Nephropathy. PKD = Polycystic Kidney Disease. RN = Reflux Nephropathy. HD = hemodialysis. PD = Peritoneal dialysis. MMF = Mycophenolate mofetil. CyA = Cyclosporine. Tac = Tacrolimus. Aza = Azathioprine. CS = corticosteroids. CCB = Calcium Chanel Blocker. MRI = Magnetic resonance imaging. eGFR = Estimated glomerular filtration rate. ERT = Estrogen replacement therapy. HC = Hypercalcemia. HPT = Hyperp arathyroidism.

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
