# Peer review of "Prevalence of Musculoskeletal Manifestations in Adult Kidney Transplant’s Recipients: A Systematic Review"

_medicina, 2021, doi:10.3390/medicina57060525_

Round 1
Reviewer 1 Report
Dear Authors,
Please write more to introduction and add more conclusions.
Please explain us about PRISMA Flow Diagram 2020. We think it is a mistake when generate it. You say that was 37 full articles included and excluded 9 articles but final you discuss only about 26 articles. What you can say about this?
Author Response
We would like to thank our reviewer for his/her nice comments and supportive review.
Please write more to introduction and add more conclusions.
author response: we added some additional comments or clarifications as necessary.
Please explain us about PRISMA Flow Diagram 2020. We think it is a mistake when generate it. You say that was 37 full articles included and excluded 9 articles but final you discuss only about 26 articles. What you can say about this?
author response: we corrected the PRISMA chart accordingly as we included a total of 28 studies (37 full articles reviewed - 9 excluded articles).
Thanks
Reviewer 2 Report
In general, a good paper, without significant faults. The authors addressed the subject in a proper way. The article is well-written and clear. The conclusions are accurate. Musculosceletal disturbances are quite common in the population of renal transplant recipients, and may be caused by numerous causes (calcium/phopspate disturbances during chronic kidney disease, immunosuppression, etc.). I think there is no novelty for the readers familiar with the subject in the article; however, the paper summarizes the evidence on the subject.
There are no mistakes in the article that should be corrected, but I feel the language can be improved to simplify reading and understanding of the text; I suggest a professional English editing service.
Author Response
In general, a good paper, without significant faults. The authors addressed the subject in a proper way. The article is well-written and clear. The conclusions are accurate. Musculosceletal disturbances are quite common in the population of renal transplant recipients, and may be caused by numerous causes (calcium/phopspate disturbances during chronic kidney disease, immunosuppression, etc.). I think there is no novelty for the readers familiar with the subject in the article; however, the paper summarizes the evidence on the subject.
There are no mistakes in the article that should be corrected, but I feel the language can be improved to simplify reading and understanding of the text; I suggest a professional English editing service.
we would like to thank our reviewer for his/her nice comments and for supporting us to publish this important review study.
The English language was reviewed by an English native-speaker colleague for accuracy and improvements. Changes are made as highlighted.